# *'once they see blood then the mood for sex is spoiled'* A qualitative exploration of female sex worker's male client views of menstruation, sex during menses and the menstrual disc

Edyth Osire[1☯], Sophie Young[2☯], Enid Awiti[1], Cynthia Akinyi[1], Fredrick Otieno[1], Penelope A. Phillips-Howard[3], Supriya D. Mehta[2,4], Linda Mason[3] *

1 Nyanza Reproductive Health Society, Kisumu, Nyanza Province, Kenya, 2 Division of Epidemiology & Biostatistics, University of Illinois Chicago School of Public Health, Chicago, Illinois, United States of America, 3 Department of Clinical Sciences, Liverpool School of Tropical Medicine, Liverpool, Merseyside, United Kingdom, 4 Department of Infectious Disease Medicine, Rush University, Chicago, Illinois, United States of America

☯ These authors contributed equally to this work.
* Linda.Mason@lstmed.ac.uk

## Abstract

To continue working during menses, female sex workers (FSW) may use unhygienic absorbents to hide their menstrual status. The menstrual disc may provide a solution. Little is known about men's knowledge and views, specifically around sex during menstruation with FSW, a population who are particularly vulnerable to violence which may be heightened during menses. To identify constructs for successful and safe menstrual disc implementation we sought to identify knowledge and attitudes towards menstruation among male clients of FSW, including views on FSW menstrual disc use during intercourse. We conducted six focus group discussions comprising a total of 51 male clients of FSW, in Kisumu, Kenya, exploring their perceptions of menses and sex during menses. In preparation for future implementation of a menstrual disc intervention, we introduced the disc to participants with an information session, answering their questions and gathering their opinions on it. Thematic analysis found most clients had limited or inaccurate knowledge about menstruation, viewing blood and the menstruator, as dirty or unclean. Sources of knowledge included school, female relatives/partners, community or church. Those reporting formal education described the biological processes more accurately, while church education lent towards stigmatized perceptions. Most participants expressed negative views towards sex during menses (i.e., unpleasurable, forbidden or risky), usually stemming from misconceptions, and reported seeking FSW services because their wife / girlfriend was menstruating. Many felt deceived if a FSW was menstruating, were often aware of materials placed vaginally to absorb blood, but generally continued with the service. No clients had prior knowledge of the menstrual disc but accepted it as a safe, hygienic, and cost-effective alternative, with some voicing interest to purchase for their wife / girlfriend. We conclude there is a need to provide accurate information on menstruation to boys and men including in school curricula and faith teaching to address lack of knowledge and negativity. These findings suggest

**Data Availability Statement:** The data is available at https://doi.org/10.25417/uic.26496877.v1.

**Funding:** This study was supported by grant number R01-A1170564 (PI: Mehta) from the National Institutes of health, National Institutes of Allergy and Infectious Diseases The funders had no role in study design, data collection and analysis, decision to publish, or preparation of the manuscript.

**Competing interests:** The authors have declared that no competing interests exist.

potential for adoption of menstrual discs by FSW with minimal adverse client reaction, and highlight possible partner support for women considering adopting a menstrual disc.

## Introduction

HIV remains a global epidemic. As reported by UNAIDS, six in every seven new HIV infections in sub-Saharan Africa (SSA) are in adolescent girls and women [1]. HIV risk is 26 times higher for women engaged in sex work [2] defined as the receipt of money or goods in exchange for sexual services [3]. In western Kenya, HIV prevalence is 12% among women in the general population and 29% in women who engage in sex work [4] Alongside challenges of increased risk of HIV and sexually transmitted infections (STIs), female sex workers (FSW) experience challenges with menstrual health and hygiene (MHH) during their work [5].

Across settings and countries, vaginal sex during menses is commonly reported by FSW. Among 1,640 FSW in Nairobi, 40% reported having sex during menses [6]. In a cross-sectional study comparing 412 FSWs to 712 general population women in Colombia, Miguez-Burbano et al. reported that 99% of FSW had sex during menses, compared to 50% of general population women [7]. In a 2008–2009 study of women employed in bars, guest houses, eating establishments or other recreational facilities in Uganda, 96% of 100 women reported transactional sex in the previous 3 months, with 41% reporting sex during menses [8].

To continue earning, some women resort to unsafe practices to hide their menses. Among workshops with 1,000 women engaged in sex work in Brazil, 42% reported inserting "cotton and other substances" (described elsewhere in the paper as "sponge" and "mattress stuffing") to "hide menses" [9]. Vaginal cleansing is also used to remove menstrual blood and other fluids produced during coitus. In a 2008 qualitative study of 30 women in Mozambique, although some women desired not to take part in sex work while menstruating, others continued taking clients due to financial needs; those who had sex during menses reported "hiding" menses from clients by washing and/or inserting a sponge into the vagina [10]. Indeed, in the study of women in Uganda engaged in sex work [8] the daily number of intravaginal cleansings was increased (p<0.01) during menses (mean 4.7 times per day) compared to non-menses (mean 4.5 times per day). Meta-analyses demonstrate that intravaginal practices, including intravaginal insertion of cloth or paper, drying or tightening agents, or cleaning with soap, are associated with increased risk of HIV, adjusting for sociodemographic and sexual exposure [11]. Among the study of FSW in Nairobi, HIV acquisition was 6-fold higher for women who had sex with clients during menses, adjusting for duration of sex work, charge per sex act, gonorrhea infection, contraceptive use, and number of unprotected sexual contacts [6]. In part this may have been due to the intravaginal practices used to hide menses from clients, though this was not evaluated in the study.

The reasons FSW take measures to hide their menses is rooted in stigma and shame, much of it driven by male clientele. Menstrual studies with men are not abundant but suggest they lack education and understanding of menstruation, which consequently perpetuates misinformation and myths, sometimes driving negative reactions. From in-depth interviews of 15 fathers in rural western Kenya aged 20 and older, it was found some first learned of menses from their wives, though school and mothers were common sources; overall accuracy and understanding of biological functions was poor [12]. Some men acknowledged the importance of knowing about menses, including whether they could have sex instead of avoiding it during this time. Their avoidance stemmed from stigma and misinformation that menstruation is

dirty and causes infection, simultaneously producing feelings of disgust. A survey in South Africa found one-third of respondents believed their partners were 'unclean' during menstruation despite the majority reporting that they were knowledgeable about menstruation [13]. Similar negative reactions were reported by peer workers (former or current FSW) whom we consulted in preparation for our wider study. As reported in numerous studies [14–16], they admitted vulnerability to violence from clients as a hazard of their job. But they told of additional susceptibility should clients discover they were menstruating. Reported reactions from clients ranged from requests (or demands) for anal sex, refusal to pay for services, to anger or force. While the limited data demonstrates men's poor knowledge of menstruation, we were unable to find studies directly assessing men's views of sex during menses in the context of sex work. The present study aimed to explore men's knowledge and attitudes towards menstruation, focusing on their views of sex during menses with FSW in order to help understand how it may contribute to unsafe or risky menstrual hygiene practices that these women undertake. It is intended this data will provide information to identify constructs for successful menstrual health program implementation for FSW.

## Materials and methods

### Ethics statement

This study was approved by the institutional review boards of Jaramogi Oginga Odinga Teaching and Referral Hospital (ISERC/JOOTRH/657/22), Rush University Medical Centre (RUMC, 22040505), University of Illinois Chicago (UIC, 2023–053) and Liverpool School of Tropical Medicine (LSTM, 22–076). Written informed consent was obtained for all participants.

The present study is a sub-study of a prospective single arm trial which assesses the impact of reusable menstrual discs on harmful MHH practices, the vaginal microbiome, Bacterial vaginosis (BV), and STIs among FSW (ClinicalTrials.gov NCT05666778; Pan Africa Clinical Trials PACTR202305912778108). This qualitative sub-study is comprised of focus group discussions (FGDs) with FSW, their male clients, and workplace-affiliated stakeholders. This paper examines the findings from the FGDs with male clients. Recruitment to the study began 08/02/2023 and ended 25/09/2023.

### Study setting

The study is set in an impoverished area, most residents live below the poverty line [17]. Kisumu is the third largest city in Kenya, yet 60% of Kisumu residents live in informal settlements where some inhabitants still embrace traditional practices such as wife inheritance, polygamy, and forced early marriage [18].The prevalence of HIV in Kisumu County is 17.5% which is over three times that of the national prevalence 4.9% [19]. In a national evaluation conducted 2017–2018 by the Kenyan Ministry of Health, the FSW population was estimated to be approximately 168,000, with 5,151 FSW estimated in Kisumu [20].

### Focus group approach

The FGD was chosen as the preferred method of data collection as we were aware of other successful studies in which men discussed menstruation through the FGD obtaining rich data. This method can facilitate a productive discussion through group dynamics, and lead to a bonding experience for participants who we hypothesised, may have felt embarrassed or reticent in talking about menstruation in a one-to-one situation with a researcher.

We originally planned to conduct four FGDs with 6 to 8 participants per group allowing for group debate while ensuring every individual could contribute. However, after four FGDs we assessed that saturation had not been achieved, particularly in relation to men's knowledge of, and information around menstruation, as well as menstrual disc insights, and an additional two FGDs were conducted.

A semi-structured FGD guide was developed to discuss topics of importance determined by study objectives, with probes to delve into relevant detail. The interview guide was developed by the lead authors, with inputs from the study team. We used the study aims as topic/ question headings, then incorporated specific questions also using the FSW interview guide where relevant and informed by the research literature where possible. The topics covered in the guide were: knowledge and attitude towards menstruation, taboos surrounding menstruation, opinions on sex during menses (including FSW vs wife), use of absorbents during sex (See S1 File). Due to future implementation of a menstrual disc intervention, we also introduced the disc during the client FGDs, providing detailed information, passing around a disc and answering their questions, and asked questions on their menstrual disc knowledge and opinions (See S2 File).

## Eligibility and recruitment of participants

FSW enrolled in our wider study were asked to identify any of their male clients who expressed interest in taking part in a group discussion on menstruation. The clients of FSW were then purposefully recruited by trained peer outreach workers who were current or former FSW. To be eligible for participation clients had to be aged 18 or older, have had a sexual experience with a FSW, be located at a sex work hotspot at the time of recruitment and consented to participate in the study. Hotspots, identified and enumerated as described in detail [21] included bars, brothels, streets, and guest houses and were chosen randomly from each peer area of coverage. The peer outreach workers identified themselves as working for a research study and provided clients with a brief explanation to the study; those who were interested provided their phone number. The study coordinator then contacted clients via telephone to confirm interest in participating in an FGD. The schedule and venue were communicated to willing participants.

## Data collection

The FGDs were conducted in a private meeting room at a local non-governmental organization (NGO) within central Kisumu town. The female project coordinator (EO) served as the moderator, assisted by a female note-taker (EA). Both were Kenyan and fluent in English, Kiswahili, and Dholuo. The coordinator or note-taker consented each participant individually in private, reading out the information sheet and consent form in the participant's preferred language. Participants were given opportunity to ask any questions. They then provided their written informed consent including for audio recording of the discussion. When the FGD was ready to begin participants were given the option to discuss in their preferred language. They were informed of 'ground rules' where they agreed that information discussed would remain confidential and sensitive issues would not be discussed outside of the FGD. Each session started with icebreakers before narrowing down to the main discussion, these lasted 1.5–2 hours in duration. The audio recordings were transcribed verbatim using a local service which employs native Kenyans who are fluent in English, Kiswahili and DhoLuo. Direct transcription was done by an independent staff and then translation was done by a different staff. This was followed by quality assurance, in which a study staff (EA or EO) listened to 10% of all interviews and compared to the transcripts and further compared the translations to the

transcripts.No names or personal identifiers were collected, and participants were represented using code (P-participant) depending on the number issued during the focus group (1–6).

## Analysis

Thematic content analysis was employed to identify important themes, generate new insights, and relate the themes to each other. This type of analysis was used as it allows the respondent's views to be organized into themes allowing the researcher to be flexible, and new themes to emerge while interpreting the responses from the participants [22]. Inductive and deductive approaches were used to reach the study aims and enable important new themes and sub-themes to emerge [23]. Transcripts were entered into NVivo pro version 12 (QSR International Ltd, Melbourne, Australia), read and re-read for familiarization with the data. After devising an initial coding framework, two research assistants (SY and EA) coded each transcript separately using an iterative process whereby new codes were added as they emerged, and any text was uncoded and reassigned as appropriate. Where there was disagreement over the interpretation or assignment of codes, the senior qualitative researcher (LM) examined the transcripts and codes and arbitrated. The coding framework was revised to reflect the agreed changes. The codes were assigned under relevant themes and subthemes with a narrative written reflecting the assigned themes and illustrated with verbatim quotes.

## Results

A total of six FGDs were held in sessions with eight to ten participants each, totaling 51 males identified as clients of FSWs. The ages of participants ranged from 20 years to 47 years, with a median age of 34 years. All participants were from Kenya, the majority (95% were of Luo (78%) or Luhya (17%) ethnicity, and 59% reported having a regular income source. Other male participants characteristics are summarized in Table 1.

## Themes

Thematic analysis revealed five key themes: (1) knowledge of menses 2) sources of knowledge, 3) attitudes towards and behaviors during sex with menstruating women, 4) communication about menses with romantic partners, FSW, and 5) perceptions of menstrual products.

**Knowledge of menses.**   Depth and Accuracy of Information—Men displayed limited knowledge when asked to define a monthly period and the menstrual process. Across groups, participants discussed feeling they lacked accurate knowledge about the menstrual cycle, and often reported feeling unsure of the complexities of menstrual health. This contributed to differences in their ability to participate and interject in the discussion, as men with limited to no knowledge often contributed less in response to questions about menses. A few participants discussed the fertilization process and were able to relate menses to reproductive functions, talking of an unfertilized egg as the catalyst for the monthly period.

*Monthly period is aah for example mostly to girls there is an age when they attain egg is formed in the fallopian tube. When it is fertilized, she will conceive if it doesn't happen, she will get her monthly period. Some call it menses. (FGD1 R7)*

A substantial number of men gave inaccurate information when asked to recall what a monthly period is. Many talked at length of dirty blood, impurities, or uncleanliness being removed from the body during the menstrual cycle. Words like "spoiled" and "dirty" were common across groups, and general sentiments seemed to reveal that male partners see menstruation as an unhygienic process.

**Table 1. Summary of male participants characteristics by focus group discussion.**

| Characteristics | FGD1 N = 10 | FGD2 N = 8 | FGD3 N = 9 | FGD4 N = 8 | FGD5 N = 8 | FGD6 N = 8 | Total N = 51 |
|---|---|---|---|---|---|---|---|
| **Age in years** | | | | | | | |
| Range | 25–45 | 24–40 | 34–43 | 21–32 | 32–47 | 20–46 | 20–47 |
| 20–29 | 2 | 2 | 0 | 6 | 0 | 3 | 13 (26%) |
| 30–39 | 6 | 5 | 7 | 2 | 4 | 3 | 27 (53%) |
| 40+ | 2 | 1 | 2 | 0 | 4 | 2 | 11 (22%) |
| Median | 36.5 | 31.5 | 35 | 24 | 39.5 | 33 | 34 |
| **Locations of Sex with FSW (Not mutually exclusive)** | | | | | | | |
| **Sex Den/ Brothel** | 8 | 2 | 4 | 3 | 7 | 6 | 30 (59%) |
| **Bar/Club/Restaurant** | 7 | 8 | 9 | 8 | 8 | 8 | 31 (61%) |
| **Lodge/Hotel/Guest House** | 7 | 7 | 8 | 8 | 5 | 6 | 41 (80%) |
| **Participant's Home** | 2 | 1 | 4 | 0 | 0 | 2 | 9 (18%) |
| **Other** | 0 | 0 | 1 | 1 | 3 | 0 | 5 (10%) |
| **Marital Status** | | | | | | | |
| **Never Married** | 2 | 2 | 1 | 1 | 0 | 2 | 8 (16%) |
| **Currently Married** | 6 | 4 | 6 | 6 | 8 | 4 | 34 (67%) |
| **Divorced/ Widowed** | 2 | 2 | 2 | 1 | 0 | 2 | 9 (18%) |
| **Schooling Completed** | | | | | | | |
| **None or less than primary school** | 2 | 1 | 1 | 2 | 0 | 4 | 10 (20%) |
| **Primary School** | 2 | 2 | 5 | 4 | 4 | 1 | 18 (35%) |
| **Secondary School** | 3 | 2 | 1 | 2 | 4 | 2 | 14 (27%) |
| **Vocational School/ College/University** | 3 | 3 | 2 | 0 | 0 | 1 | 9 (18%) |

*Monthly period is a process where ladies or women goes during their monthly period to remove the dirty clots that are inside their system so that they can have a clear system. (FGD6 R7)*

When asked the ages at which girls enter into puberty, several men seemed to suggest that the ages vary depending on the individual's hormones or biology, but commonly reported between 9–12. A few men reported that menstruation started later, around 17–18 years.

*I think that it depends and varies per individual, some start as early as 9 years and mostly it is 12–13 years, there are those who start experiencing it early. (FGD2 R3)*

**Sources of knowledge.** Among men who were able to vocalize their perceptions and ideas about menstruation, the reported sources of their information were variable. Men generally seemed unsure of where their information came from; indeed, one participant stated that his information came from his own mind. However, several men discussed learning about women's reproductive health through their formal education.

*With me, most of us got an opportunity to go to school. So, with me, I started knowing about periods while I was in school. I was taught about periods, what is it, how period it takes. Thereafter I engaged with girls in the college, I could notice what was happening. (FGD6 R7)*

*My opinion is much closer to what my colleague has just said, that the reproductive system we started learning from primary from a subject called science. That is where we learnt those parts and how those processes happen. Again when we reached high school we found it in biology and again we learned about it in detail. (FGD3 R4)*

A few men also reported gaining knowledge directly from the women in their lives namely romantic and sexual partners. However, they also discussed talking to sisters and female friends about menstruation. Some indicated that their information came from their communities, passed along between households and across social groups by word of mouth. Several men also reported learning their information from the teachings and community of their church. The accuracy and depth of knowledge acquired by men varied across sources, with those reporting formal education describing more accurately the biological processes, and church education incorporating more stigmatized information. Many of the men cited church and faith-based education as a source of information regarding menses, and these religious beliefs were often illustrated throughout these discussions.

*The reason why I say so is that I am so sure about it. It was preached to us at the church, that any woman who is on her menses is a dirty person and she is limited to doing so many things. (FGD3 R1)*

**Attitudes and behaviour.** Attitudes Towards Sex During Menses—The majority of respondents expressed negative views towards sex during menses, suggesting it is unpleasurable or forbidden, and this was apparent across all six FGDs. Men described these feelings as either rooted in their faith, or its impact on their sexual pleasure.

*It is said biblically when a woman is on menstruation, they are unclean blood, you should not have an intercourse with her because she is like a contaminated person you should not engage with biologically. (FGD2 R5)*

*It is not sweet anymore because it is like you have entered into mud. (FGD6 R3)*

Attitudes that were ambivalent or positive towards having sex during menses were uncommon. A few men expressed that they found no problem engaging, while others expressed that some men view menstruation as a desirable time to have sex, either due to sexual preferences or for family planning purposes.

*Okay according to me, I don't see any problem if you have sex with someone on her menses. Unless she has a problem, she can be excused but if there isn't a problem then one could just proceed. (FGD3 R7)*

Several men also discussed their ideas about the risks of having sex with menstruating women, many of which were rooted in misconceptions or myths. For instance, many men expressed fears that they could give or get "Pelvic Inflammatory Disease", contract "dangerous" STIs from "dirty" menstrual blood.

*When I have sex with you when you are menstruating maybe that will not affect you but when a man like me speaks and that has ever happened to him, something like pelvic inflammatory disease, you might force a lady when she is menstruating to have sex with you, but 2 weeks cannot lapse before you get pelvic inflammatory disease. (FGD4 R3)*

Sexual Practices During Menses—Several men across different groups reported a complete aversion to engaging in sex with a wife or romantic partner during their menstrual period, often stating this as a reason they seek the services of an FSW. Others reported reduced sexual interest.

*If the main chick is menstruating, you go to plan B. (FGD4 R3)*

*What I can add again is that sex is all about pleasure you have for fun when you are having sex. And the moment one is unclean, and if you want to have sex you will not enjoy it because you will be doing it unwillingly. So, one has to calm down until the menses are over then you will enjoy sex afterwards. (FGD1 R8)*

However, a few men reported maintaining a normal sex life with their wives or girlfriends who are menstruating, claiming a responsibility to fulfill their partners' needs.

*Some women who are horny when they are attending and when we are married, she has conjugal rights objective, am the one to assist her and this exceeds her only when she is attending, and she is horny and needs my attention. (FGD2 R5)*

In contrast, discussions concerning sex workers tended to consist of a don't ask, don't tell attitude towards sex during menses. Multiple men reported that upon realizing an FSW is menstruating, they feel deceived but continue engaging anyway to relieve their sexual urges. Alternatively, a few men reported that they might ask for their money back and leave without having sex at all.

*This is because we consider her impure during this period, but since I have this uncontained sexual urge, I would call this woman and have sex with her. I would call her that so and so today I would want to have you, and you understand it is only a secret between how many? (FGD5 R3)*

Men discussed at length about the differences in their physical behavior when engaging with a menstruating romantic partner versus a FSW. Participants tended to report a more utilitarian approach to sex with FSW, opting to engage in the act quickly and move about their day. However, they also discussed an elevated willingness to use protection with sex workers, lending them to feel more protected against blood-borne infections when having sex with them.

*And again the difference with one, the sex worker you know they don't always accept free intercourse, you understand the one where you have sex without protection, you can have unprotected sex with your spouse or wife. so you will find that as man if such a thing happens you will find that, if it is a sex worker, it will be inside a ballon (condom)so some will decide just to finish because there will be, he sees that he is a bit safe (M: Protection?) and it will have protection, so others will decide to finish because when you remove it the condom is what will have the blood which is not yours (FGD6 R7)*

When it came to wives or partners, participants reported performing other acts of intimacy instead, indicating that menstruation more obviously interrupts sex with their partners.

*Between your wife or sexual partner that is why they do it like that in the sex den, you come in have sex and go away but for your wife you have to prepare her and do foreplay then you go to the main process that is sex. So, there is a little difference. (FGD1 R7)*

Alternatives to vaginal sex with a menstruating partner were also detailed by a few of the participants, and they reported engaging in anal sex or masturbation. One participant reported engaging in sex with other men to indulge their sexual desires.

*Someone may see the vagina as having blood then he can have anal sex. (FGD1 R7)*

*Sometimes I could do the wanking or rather have a boyfriend outside. (FGD3 R1)*

**Communication about menstruation.** With FSW

- Men discussed their feelings towards FSW communicating with them about their menses. Participants seemed split in their desire for disclosure of a menstrual period. Several men indicated that they would like to be told when a sex worker is menstruating, either so they could find a different partner or adjust their expectations. A few men discussed favoring communication so that they might avoid contracting infections.

*She has to inform you that she is in her menses, it is good for the sex worker to tell you today I am on my menses it will not be possible maybe let's have some lunch maybe supper and you part you go about your activities I think that can be good. (FGD1 R8)*

In contrast, several other men suggested they prefer not to be told when a sex worker is on her menses, as it interferes with the sexual gratification.

*I don't think she should tell you because when she tells you, I will still be just the same point that I was before because when she tells you it is like you have gone out to meet with her then she tells you that she is in her period, and when she is telling you that she is menstruating you haven't even gone to a room, what will come to mind. . . I will just leave and look for someone else. (FGD4 R1)*

With Wives or Partners -Compared to their discussions in relation to FSW, men discussed at length their expectations for communication with their romantic partners. In general, men seemed to favor open communication and disclosure with a wife or girlfriend, wanting to be told when their partner is menstruating. Many discussed the importance of this information in terms of family planning and avoiding pregnancy. However, other men suggested their motivations came more from wanting the ability to visit a FSW if their romantic partner is menstruating. They suggested that open communication allows them to plan ahead in these instances.

*You know a man must know when a woman is menstruating, they should know because you know a night can be long for a man, because the night is long sometimes you touch her and you are told no, so you must be told about it in good time that the lady is menstruating because something might pass before in during the day and he tells himself no, let me just go and finish up at home, but when he gets home he finds a block there so a man must know. (FGD4 R3)*

*It is required of us to have the information on menstruation so that we can practice family planning within our houses. We could also understand the safe days to have sex and unsafe days. As men it is very important that we have that information. (FGD5 R6)*

**Products.** Payment, Cost, and Accessibility -Prior to our FGDs, many of the male participants were unaware of the various products women use to manage their menses, although it appeared they had some prior knowledge of sanitary pads. Participants were asked how much they think various menstrual products cost and displayed largely accurate knowledge: most reported the cost of a pack of pads as ranging from 50 to 200 Kenyan shillings (KSH) ($0.25–1.50 USD; actual costs range from $0.32–1 per pack depending on quality). However,

participants seemed to reach consensus on the unaffordability of purchasing these products repeatedly over time, expressing sentiments that women often can't afford to comfortably manage their menstrual periods.

> It is a loss, there are some people who are challenged financially, maybe they can't raise that amount of money to buy those things. (FGD2 R10)

Despite agreeing on the high costs of menstrual hygiene products, men seemed to differ in their opinions of who should be responsible for purchasing the products. Several respondents suggested that men should purchase sanitary towels for their partner, while others felt that women are best positioned to purchase such products for themselves. A few participants discussed the intervention of NGOs or the Kenyan government to subsidize the costs of products.

> Personally, what I can say is that as men, if you know that your wife is on her menses then you should buy those things that they use and bring to her for use, yes. So that she could maintain hygiene, yes. (FGD3 R2)

> In my opinion, women should buy this themselves, this is because at times she could keep it somewhere and get you surprised and asking. "Yes! What is this?" And that she had bought it long ago and she just kept it. So I believe it is the woman who should buy it. (FGD5 R3)

Many men reported sentiments against the use of certain products commonly used in Kenya, especially by FSW, to block blood flow during sex. Men seemed to agree that the use of products such as cotton wool and tissue inside of the vagina is unhygienic and detrimental to the health of women.

> I think those girls should not use tissue paper; they are made from many chemicals. So, when they stuff the inside, I think they introduce some infections within them which they cannot detect at the early stages. So, they should be told not to use tissue paper. (FGD5 R8)

> What I know well is that tissue is not something to insert inside given how I know tissue, it is something that if you fold and place it there and we all know that the place is usually wet and when you place tissue into it, the tissue easily torn into pieces such that by the time you will be removing it, it would have gone deep inside her and that is what is going to give her an infection. (FGD2 R6)

**Menstrual discs.**   Prior to our introducing men to the menstrual disc none had any knowledge of it. Men were shown a reusable menstrual disc which could be worn during sex. The moderator reviewed a graphic that showed where the menstrual disc sits within the vaginal canal. They were allowed to handle the disc (it was passed around by the group members), and were given information on its cost, use, cleaning, durability, and safety profile. Following the introduction of the menstrual disc, a majority of men appeared open-minded and seemed to hold positive sentiments towards women using them. They generally accepted that the menstrual discs were a safe, hygienic, and cost-effective alternative to other products. In particular, common responses discussed menstrual discs as a desirable replacement for the use of cotton wool or tissue. Respondents also expressed feelings that use by FSW during menstruation could improve their sexual experiences.

> For me, what I think is, I think when dealing with menstruation it [the menstrual disc] is a big YES. Another thing also is when we come to financial, it is economical and I think

*hygienically it is one of the best, if you can keep it clean for future use, I think it is effective. (FGD1 R7)*

*I think this could be even better than sanitary pads, yes. Even if I buy it for a sex worker in town or even my wife it can be prevented. But then there are some men who are scared away or shy away easily, once they see blood then the mood for sex is spoiled. But when he doesn't see that blood, he could have the morale with his wife. (FGD3 R9)*

Questions prompted by the moderator also focused on whether or not sex workers should disclose the use of a menstrual disc were they to use one. Several men indicated that they would prefer to be told and suggested those feelings arose from personal curiosity. Other men spoke of wanting to be made aware so that they might trial the menstrual disc with a sex worker, as they may later want to purchase for a wife or girlfriend.

*For me I would wish because when I have known more about it, I would want to know how it affects her, how she reacts when she is open. (FGD2 R3)*

However, other men stated that they wished to remain ignorant to the menstrual cup's presence, as it wouldn't interfere with sex and knowing might affect their sexual appetite.

*I don't think it is that necessary because this man comes to seek services. The service that you offer to him will not result in you seeing stains. They should just keep it within themselves. (FGD5 R8)*

*Even me, I do not want to know because if I do, I will lose my appetite. (FGD6 R5).*

## Discussion

Economic hardship often necessitates FSW taking clients while they are menstruating, resorting to unhygienic and potentially hazardous materials and practices to absorb and hide blood flow. In preparation for trialing a menstrual disc that can be worn during sex in a sample of FSW we sought to assess whether there may be risk from negative or hostile client reaction to disc use. Delving into the perspectives around menstruation and menstrual sex with clients of FSW, whose views on menstruation have not–to the best of our knowledge–been sought, we found many of these male participants had limited and/or inaccurate knowledge about menstruation. Negative attitudes towards menses and the menstruating woman were coupled with a lack of desire to have sex with a girlfriend or spouse who was menstruating. Some participants reported resorting to the services of a FSW for this reason, with less regard to sex workers menstrual status. Yet, our participants appeared curious and receptive to the menstrual disc. We consider implications from their narratives below.

A few of our participants appeared knowledgeable about menstruation, although the majority lacked accurate or in-depth knowledge, with some aware that they had limited knowledge. This adds to findings from the small but growing body of studies that highlight men's ignorance of this topic [24–26]. Our analysis found a tendency for men displaying accurate or greater knowledge to report having received this education from school. That these were in a minority may be partly explained by our participants' schooling level, with over half receiving primary education only, and nearly one fifth not attending school at all. Studies have also highlighted that school education can misinform [27]. In Tanzania a high proportion of boys received information at school, although 20% still reported that '*periods are unnatural*' [25] in India [28] despite being on the curricula, puberty and menstruation were not always taught.

Similar to other studies [29–31], many of our participants learnt about menstruation in a fragmented way via female partners, relatives and community groups. We noted a common source of misinformation was attributed to faith teaching, which highlighted harmful attitudes towards menstruation and the menstruating woman. In our study the Bible was quoted; more widely, other religions including Judaism and Hinduism amongst others similarly are considered to contribute to taboos preventing women from participating in society, in religion and even in their own home during menses [32,33].

As discussed by Erchull [26], lack of information and misinformation combine to shape negative attitudes toward menses. The literature is abundant with both male and female accounts of menstruation being 'shameful', menstrual blood being labelled 'dirty' and 'polluting' and menstruating women 'impure' [29,33]. Some of our participants echoed these accounts, as illustrated by our exemplar quote '*any woman who is on her menses is a dirty person*'. Although this view was not shared by all, it highlights the need for factual unbiased teaching.

Few studies have considered men's engagement with, and perceptions about having sex during menstruation. There are some risks associated with having sex during menstruation. There is a higher risk of transmission of blood borne viruses such as HIV, whilst the changing alkalinity of the vagina, conducive for microbes to flourish [34] may contributed to increased risk of STI acquisition [35] These risks were identified, albeit in a rudimentary way, by a minority of our participants. However, others voiced that *they* could develop or contract pelvic inflammatory disease (PID; which affects women), a misconception that we have noted in our other field studies in western Kenya. During this study, and our other field work, men quite commonly spoke of having PID, or being at risk of contracting PID, rather than transmitting the infectious organisms that may lead their female partner to develop PID. Because of this repeated misconception, further understanding of men's perceptions and knowledge of RTI's and STI's and sequelae, including PID, is warranted within the study setting, particularly considering their apparent lack of knowledge generally around reproduction.

Given lack of knowledge and prevailing negative attitudes toward menstruation, it is not surprising that many of our participants had negative perceptions about menstrual sex, ranging from it being merely undesirable to outright disgust. Others voiced that their faith taught them that sex during menstruation is forbidden, once again demonstrating the negative influence of faith teaching. There is very little research to draw upon in which to make comparison, although from the few studies conducted, it seems some report the messiness of menstrual sex combined with feelings of disgust for menstrual blood itself drives a prevalent hostile reaction [29]. Engaging in menstrual sex is also seen as shameful behaviour [36]. However, in one study women reported that men most pursued them for sex when they were menstruating [33] whilst participants in Macleod et al's study [37] who made derogatory comments about menstruating women, described using self-control to refrain from sex at this time, rather than not desiring it. In another study [36] some men, more typically those who were older and more sexually experienced and in committed relationships, described menstrual sex as part of a trusting and intimate relationship. We similarly noted that some of our participants discussed having sex during menstruation within the context of their normal sex life with a wife or girlfriend. But more commonly, many of our participants reported their wife or girlfriend having their menses was a key reason for seeking the services of a FSW. Like Perenovic and Bentley [24] we do not know which partner was key in choosing not to have menstrual sex (i.e., the man or the wife / girlfriend), or indeed whether it was by mutual agreement. Notably, if a FSW might be menstruating, this appeared less of a problem for our participants. Unlike a partner who may dislike or refuse to have menstrual sex, the FSW continues her business and is thereby available for sex. According to our conversations with sex workers, and the few studies

conducted [9,10] FSW often hide their menstrual blood when taking clients who are thereby less likely to be aware of their status, or less troubled by the sight or amount of menstrual blood. Although no participants admitted a violent reaction on finding out that a FSW was menstruating, a phenomenon that our peer workers had told us sometimes occurred, they did admit feeling deceived, asking for their money back, leaving without using that FSW services, and possibly turning to another FSW.

To our knowledge, this paper is the first to consider men's views of the menstrual disc. Acknowledging that men's views were gathered immediately after they had received an interactive information session, we are aware that views may have been biased. However, we are encouraged by the curiosity and receptivity of the men, apparent across all FGDs. We believe this potentially bodes well for FSW adoption without fear of reprisals should their clients be aware that they are using it. Receptivity was further demonstrated by the number of participants expressing interest in obtaining a disc for their wives or girlfriends. Whilst giving reason for optimism, we also note a potential drawback. Studies show that many girls or women are comfortable using a menstrual cup, but others choose not to try or give up before they have mastered its use [38], and like all forms of menstrual hygiene, usage should be a matter of personal choice. Choosing to abstain from sex during menses may not stem from issues around blood, but rather from cultural or faith beliefs, physical discomfort or dysmenorrhoea [39]

Our study is set in Kenya where cultural mores, such as patriarchy and legalized polygamy, act to constrain women's independence, with high rates of interpersonal violence (IPV) and sexual harassment [17,40], and constrained autonomy in AGYW's sexual relationships [17]. Within the present study, we note some views expressed by our participants may imply this male power dominance–'*you might force a lady when she is menstruating to have sex with you*'; and '*Unless she has a problem, she can be excused but if there isn't a problem the one could just proceed*'. Against this background we therefore highlight concerns that male involvement in obtaining or encouraging their partner to use a menstrual disc should not be motivated by masculine sexual self-interest but should consider and respect their partner's needs and wishes.

We acknowledge limitations of our study that include use of the FGD to obtain men's views. The FGD successfully garnered rich discussion around a taboo topic, it may be that our participants leant towards a more hegemonic masculinity in their communication with other men, and about a topic historically seen as 'women only' business. However, we noted findings were similar across FGDs, with a range of knowledge and views displayed within each group. The moderator and note taker were Kenyan females. This may have acted to temper stereotypical views or further elicit such views. Our choice to use a female moderator stems from her previous experience and ability to engage men in a discussion, which has been successful in opening a lively and rich dialogue in the present study. The menstrual disc used in our information sessions is a type that is conducive to wearing during intercourse. Other types of menstrual disc may produce a different and perhaps less enthusiastic response amongst men.

## Conclusion

Our study has added to the limited research on men's knowledge and attitudes toward menstruation focusing on their views of sex during menses with FSW to help understand how it may contribute to unsafe or risky menstrual hygiene practices that these women undertake. This research highlights men's lack of knowledge and largely negative perceptions about the topic, including towards sex during menses. Researchers, policy makers, politicians alike have been calling for increased awareness of boys and men around menstruation. We echo this need, calling for accurate information including on school curricula, and through faith

teaching where appropriate, to address the current lack of awareness that contributes to stigma and shame, and perpetuation of menstrual taboos. This research illuminates men's need for education around reproductive health in general, noting that participants were receptive to, and curious about a reusable menstrual disc that can be worn during sex, which suggests that this has potential to be adopted by FSW with minimal adverse client reaction. Further it highlights possible partner support for women considering adopting a menstrual disc in the future. These study findings can be used to inform successful menstrual health programme implementation for FSW in the future.

## Supporting information

**S1 File. Community men FGD guide.**
(PDF)

**S2 File. Menstruation script.**
(PDF)

## Acknowledgments

We thank the following Peer Leaders for their assistance in identifying and referring participants for this study (in alphabetical order): Mwanaisha Achieng, Ann Amolo, Mildred Anyango, Eunice Chuch, Jecinta Jung'a, Millicent Ochieng, Lillian Odeyo, and Rose Oyugi.

## Author Contributions

**Conceptualization:** Penelope A. Phillips-Howard, Supriya D. Mehta, Linda Mason.

**Data curation:** Edyth Osire, Sophie Young, Enid Awiti, Cynthia Akinyi.

**Formal analysis:** Sophie Young, Supriya D. Mehta, Linda Mason.

**Funding acquisition:** Penelope A. Phillips-Howard, Supriya D. Mehta.

**Investigation:** Edyth Osire, Sophie Young, Enid Awiti, Cynthia Akinyi, Supriya D. Mehta, Linda Mason.

**Methodology:** Fredrick Otieno, Penelope A. Phillips-Howard, Supriya D. Mehta, Linda Mason.

**Project administration:** Fredrick Otieno, Supriya D. Mehta.

**Supervision:** Fredrick Otieno, Supriya D. Mehta, Linda Mason.

**Validation:** Sophie Young, Supriya D. Mehta, Linda Mason.

**Visualization:** Sophie Young, Supriya D. Mehta, Linda Mason.

**Writing – original draft:** Edyth Osire, Sophie Young, Enid Awiti, Cynthia Akinyi, Supriya D. Mehta, Linda Mason.

**Writing – review & editing:** Edyth Osire, Sophie Young, Enid Awiti, Cynthia Akinyi, Fredrick Otieno, Penelope A. Phillips-Howard, Supriya D. Mehta.

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
