## [Decision Letter · Decision Letter 0]

25 Jul 2024

PONE-D-24-10017'Once they see blood then the mood for sex is spoiled’ A Qualitative Exploration of Female Sex Worker’s Male Client Views of Menstruation, Sex during Menses and The Menstrual DiscPLOS ONE

Dear Dr. Mason,

Thank you for submitting your manuscript to PLOS ONE. After careful consideration, we feel that it has merit but does not fully meet PLOS ONE’s publication criteria as it currently stands. Therefore, we invite you to submit a revised version of the manuscript that addresses the points raised during the review process.

In your revision please carefully address the issues raised by the two reviewers especially the issue of reviewer 2 on ethics of using focus groups for such a sensitive population. Provide a solid justfication why you used the approach I also invite you to carefully address the concerns raised by reviewer one paying particualr attention to linking the introduction and discussion to the research gap. 

We look forward to receiving your revised manuscript.

Kind regards,

Martin Mbonye

Academic Editor

PLOS ONE

Reviewers' comments:

Reviewer's Responses to Questions

**Comments to the Author**

1. Is the manuscript technically sound, and do the data support the conclusions?

Reviewer #1: Yes

Reviewer #2: No

2. Has the statistical analysis been performed appropriately and rigorously? 

Reviewer #1: N/A

Reviewer #2: N/A

3. Have the authors made all data underlying the findings in their manuscript fully available?

Reviewer #1: Yes

Reviewer #2: No

4. Is the manuscript presented in an intelligible fashion and written in standard English?

Reviewer #1: Yes

Reviewer #2: Yes

5. Review Comments to the Author

Reviewer #1: I thank the authors for a well-presented paper. It clearly shows that they have done a lot of work following well-executed research. Overall the paper is presented well. I however have some suggestions to make it even better and more significant for the field.

Major issues

Introduction

The research gap that the authors seek to fill does not come out well. The authors mention men’s poor knowledge of menstruation, and also mention the lack of information on men’s views on sex during menses, but do not link these to gaps in literature that need to be filled. The authors could consider adding links between the above-mentioned items with health/well-being outcomes for women or men.

Minor issues

Abstract

- The gap the research is seeking to fill does not come out clearly. Edit lines 21-28 to clarify.

- The conclusion is not clear is not clear (lines 37-39).

Introduction

- Line 48 is missing punctuation.

- Add citation to statement lines 48-49.

- Authors could consider rewriting line 57 beginning with ‘To continue earning…”

- Line 85: attitudes.

Methods

- The first sentence seems irrelevant to the study (lines 97-99).

- Punctuation: lines 101, 104.

- Line 106: consider changing language from “we aimed to..” to “we originally planned to…” Related to this, change instances of similar language in the paper. These are things the authors have carried out, so the language should reflect this.

- Line 110: mention how the interview guide was developed.

- Line 119: Consider rephrasing to “Regular and casual..” Give details on how the men were recruited. Did it matter that they were regular or casual, and was any care given to balance the numbers of regular vs casual partners?

- Line 129: consider changing to “conducted in..”

- Line 133-136: the authors need to mention of participant information and consent process.

- Line 136-137: give details of the local transcription service.

Results

- Line 159: review the percentages.

- Table 1: add lower border.

- Line 165: consider separating knowledge and sources of knowledge as themes. This will give you the freedom to explore them better in the results and discussion.

- Line 181: Consider using another word instead of ‘however’. Using ‘However’ proposes a contradiction with the previous statement, yet there is no contradiction.

Discussion

- Lines 384: This point is a good and strong one that doesn’t come out in the introduction. One suggestion I have is that the authors could consider using this point (possibility of negative or hostile client reaction due to disk use) as a foundation to build the paper to avoid or reduce violence against women who use the disc use. This way, violence reduction could be the significance of the paper. This is one suggestion to illustrate how to tie the aim and significance to outcomes for women. The authors should think through as well what other outcomes they can consider and incorporate into the introduction, results and discussion.

Reviewer #2: I reviewed this paper carefully. I believe that conducting a research that is high sensitive, using Focus Group Discussion, in a place that is organized and let the participants share their experience together is unethical.

6. PLOS authors have the option to publish the peer review history of their article (what does this mean?). If published, this will include your full peer review and any attached files.

Reviewer #1: No

Reviewer #2: No

---

## [Author Response · Author response to Decision Letter 0]

19 Aug 2024

Dear Reviewers,

Thank you for taking the time and effort to review out paper. We have outlined below the changes we have made in accordance with the suggestions made and feel that the paper has been strengthened by the constructive feedback. We hope that our amendments meet with your standards and that the paper is ready to be accepted for publication.

Reviewer #1: I thank the authors for a well-presented paper. It clearly shows that they have done a lot of work following well-executed research. Overall the paper is presented well. I however have some suggestions to make it even better and more significant for the field.

The authors would like to thank you for your positive and constructive feedback. 

Major issues

Introduction

The research gap that the authors seek to fill does not come out well. The authors mention men’s poor knowledge of menstruation, and also mention the lack of information on men’s views on sex during menses, but do not link these to gaps in literature that need to be filled. The authors could consider adding links between the above-mentioned items with health/well-being outcomes for women or men.

Thank you for reflecting on this point. We agree and have pulled out further information on the important issue of violence as suggested in your remarks on the discussion. (Minor word changes have also been incorporated throughout the paper to also highlight this issue).

The introduction now reads:

‘The reasons FSW take measures to hide their menses is rooted in stigma and shame, much of it driven by male clientele. Menstrual studies with men are not abundant but suggest they lack education and understanding of menstruation, which consequently perpetuates misinformation and myths, sometimes driving negative reactions. From in-depth interviews of 15 fathers in rural western Kenya aged 20 and older, it was found some first learned of menses from their wives, though school and mothers were common sources; overall accuracy and understanding of biological functions was poor (12). Some men acknowledged the importance of knowing about menses, including whether they could have sex instead of avoiding it during this time. Their avoidance stemmed from stigma and misinformation that menstruation is dirty and causes infection, simultaneously producing feelings of disgust. A survey in South Africa found one-third of respondents believed their partners were ‘unclean’ during menstruation despite the majority reporting that they were knowledgeable about menstruation (13). Similar negative reactions were reported by peer workers (former or current FSW) whom we consulted in preparation for our wider study. As reported in numerous studies (Bhattacharjett et al 2018; Jewkes et al 2021; Spyrelis and Ibisomi 2022), they admitted vulnerability to violence from clients as a hazard of their job. But they told of additional susceptibility should clients discover they were menstruating. Reported reactions from clients ranged from requests (or demands) for anal sex, refusal to pay for services, to anger or force’. 

Minor issues

Abstract

- The gap the research is seeking to fill does not come out clearly. Edit lines 21-28 to clarify.

We have amended to read as:

‘Little is known about mens knowledge and views, specifically around sex during menstruation with a FSW, a population who are particularly vulnerable to violence which may be heightened at such times. To identify constructs for successful and safe menstrual disc implementation we sought to identify knowledge and attitudes towards menstruation among male clients of FSW, including views on FSW menstrual disc use during intercourse’.

- The conclusion is not clear is not clear (lines 37-39).

We have rewritten this as follows:

‘We conclude there is a need to provide accurate information on menstruation to boys and men including within the school curricula to address male ignorance and negativity. When faith teaching covers this topic, it needs to be fact based’.

Introduction

- Line 48 is missing punctuation.

Addressed

- Add citation to statement lines 48-49.

This has now been added. 

Phillips-Howard P, Osire E, Akinyi C, Zulaika G, Otieno FO, Mehta SD. Water, sanitation and hygiene at sex work venues to support menstrual needs. 2024; 

- Authors could consider rewriting line 57 beginning with ‘To continue earning…”

This has been changed to

‘To continue earning some women resort to unsafe practices to hide their menses’.

- Line 85: attitudes.

Amended

Methods

- The first sentence seems irrelevant to the study (lines 97-99).

We have removed this line.

- Punctuation: lines 101, 104.

Amended

- Line 106: consider changing language from “we aimed to..” to “we originally planned to…” Related to this, change instances of similar language in the paper. These are things the authors have carried out, so the language should reflect this.

We have addressed the specific line and have also amended the paper generally where we found errors in the tense used. 

- Line 110: mention how the interview guide was developed.

We have included the following.

‘The interview guide was developed by the lead authors, with inputs from the study team. We used the study aims as topic/question headings, then incorporated specific questions also using the FSW interview guide where relevant and informed by the research literature where possible’. 

- Line 119: Consider rephrasing to “Regular and casual..” Give details on how the men were recruited. Did it matter that they were regular or casual, and was any care given to balance the numbers of regular vs casual partners?

Thank you, this is a valid point. We had assumed it was important to let the reader know that both regular and casual clients were eligible to participate in this study. On reflection, we did not target recruitment, nor analyse according to these categories. We have therefore removed mention of ‘regular’ or ‘casual’ just mentioning that they were male clients of FSW. 

- Line 129: consider changing to “conducted in..”

Amended

- Line 133-136: the authors need to mention of participant information and consent process.

Thank you for the suggestion. We have provided the additional details:

‘The FGDs were conducted in a private meeting room at a local non-governmental organization (NGO) within central Kisumu town. The female project coordinator (EO) served as the moderator, assisted by a female note-taker (EA). Both were Kenyan and fluent in English, Kiswahili, and Dholuo. The coordinator or note-taker consented each participant individually, and in private, reading out the information sheet and consent form in the participants preferred language. Participants were given opportunity to ask any questions. They then provided their written informed consent including for audio recording of the discussion’.

- Line 136-137: give details of the local transcription service.

We have included the following;

‘The audio recordings were transcribed verbatim using a local transcription service which employs the use of local workers who are fluent in English, Kiswahili and DhoLuo. Direct transcription is done by an independent staff and then translation is done by a different staff. The audio, transcription and translation are then submitted for an inhouse quality assurance by a study staff. The inhouse quality assurance involves listening to 10% of all interviews and comparing to the transcripts and further comparing the translations to the transcripts’.

Results

- Line 159: review the percentages.

Amended

- Table 1: add lower border.

Amended

- Line 165: consider separating knowledge and sources of knowledge as themes. This will give you the freedom to explore them better in the results and discussion.

We have taken your suggestion and amended the results to reflect there were 5 key themes, having separated knowledge out from sources of menstruation.

- Line 181: Consider using another word instead of ‘however’. Using ‘However’ proposes a contradiction with the previous statement, yet there is no contradiction.

Amended

Discussion

- Lines 384: This point is a good and strong one that doesn’t come out in the introduction. One suggestion I have is that the authors could consider using this point (possibility of negative or hostile client reaction due to disk use) as a foundation to build the paper to avoid or reduce violence against women who use the disc use. This way, violence reduction could be the significance of the paper. This is one suggestion to illustrate how to tie the aim and significance to outcomes for women. The authors should think through as well what other outcomes they can consider and incorporate into the introduction, results and discussion.

We thank reviewer 1 for this constructive feedback and agree that this information is needed in the introduction to enhance the thread running through the paper. We have addressed within the introduction (see above) and have also strengthened corresponding sections within the results and discussion. This includes the following:

‘Although no participants admitted a violent reaction on finding out that a FSW was menstruating, a phenomena that our peer workers had told us sometimes occurred, they did admit feeling deceived, asking for their money back, leaving without using the FSW services or possibly turning to another FSW’. 

Reviewer #2: I reviewed this paper carefully. I believe that conducting a research that is high sensitive, using Focus Group Discussion, in a place that is organized and let the participants share their experience together is unethical.

Thank you for your careful review of our paper. We hope to reassure you of the ethicality of our study with the following information, some of which we have also inserted into the methods section (see below) so that the reader can also be reassured. We feel our paper has been strengthened in response to your feedback. 

We have provided details of the four ethical review boards that gave approval for our study in the methods section, as requested by the Editor.

‘This study was approved by the institutional review boards of Jaramogi Oginga Odinga Teaching and Referral Hospital (ISERC/JOOTRH/657/22), Rush University Medical Centre (RUMC, 22040505), University of Illinois Chicago (UIC, 2023-053) and Liverpool School of Tropical Medicine (LSTM, 22-076). Written informed consent was obtained for all participants’.

We have provided further detail of the consenting process as requested by reviewer 1.

‘The coordinator or note-taker consented each participant individually, and in private, reading out the information sheet and consent form in their preferred language. Participants then provided their written informed consent including for audio recording of the discussion’.

We have provided further detail in the methods of why we chose the focus group discussion as our method of data collection. 

‘The FGD was chosen as the preferred method of data collection as we were aware of other successful studies in which men discussed menstruation through the FGD obtaining rich data (Mohamed et al 2018; MacLeod et al 2023 ). This method allows group dynamics, and a bonding experience for participants who we hypothesised, may have felt embarrassed or reticent in talking about menstruation in a one to one situation with a researcher’.

We feel that this decision was vindicated by the collection of rich data, with participants opening up and sharing their thoughts with the group. (A number of them on leaving took the time to tell the moderator they had valued the experience, particularly the education section, and would like to take part in another discussion). 

We have included further detail of the recruitment process which illustrates that there were 2 points of contact (first with their sex worker and then a peer support worker) prior to potential participants having – with their agreement - their contact details passed to our study coordinator. This affords the potential participant sufficient opportunity to express lack of interest in participation without having their details passed on to our study co-ordinator. The study co-ordinator contacted interested participants providing further detailed information and explored whether each participant was still willing to participate, prior to giving details of the time and venue following a positive response. Please note, participants turned up of their own accord to the venue on the day of their FGD. Their transport costs were reimbursed, refreshments were provided, but they were not paid for their time.

‘FSW enrolled in our wider study, were asked to identify any of their male clients who expressed initial interested in taking part in a group discussion on menstruation. The clients of FSW were then purposefully recruited by trained peer outreach workers who were current or former FSW. To be eligible for participation clients must be aged 18+, have had a sexual experience with a FSW, be located at a sex work hotspot at the time of recruitment and consented to participate in the study. Hotspots, identified and enumerated as described in detail (18) included bars, brothels, streets, and guest houses and were chosen randomly from each peer area of coverage. The peer outreach workers identified themselves as working for a research study and provided clients with a brief explanation to the study; those who were interested provided their phone number. The study co-ordinator then contacted clients via telephone to confirm interest in participating in an FGD. The schedule and venue were communicated to willing participants. 

We have provided further detail in the data collection section on the consent process. Please note consent was done in private and included specific consent for audio recording. Our consent form states clearly that any participant can change their mind and withdraw at any point without penalty. We also incorporated further participant safeguards by introducing ground rules to remind all participants of the need for confidentiality and non-disclosure outside of the venue. Participants were also given opportunity to set any additional ground rules they would like to have. Anonymity was maintained in the discussion with the use of participant numbers. No names were used. 

‘The FGDs were conducted in a private meeting room at a local non-governmental organization (NGO) within central Kisumu town. The female project coordinator (EO) served as the moderator, assisted by a female note-taker (EA). Both were Kenyan and fluent in English, Kiswahili, and Dholuo. The coordinator or note-taker consented each participant individually, and in private, reading out the information sheet and consent form in their preferred language. Participants then provided their written informed consent including for audio recording of the discussion. When the FGD was ready to begin participants were given the option to discuss in their preferred language. They were informed of ‘ground rules’ where they agreed that information discussed would remain confidential and sensitive issues would not be discussed outside of the FGD’.

In addition to the 2 references now included in the paper, we have also included below a short list of additional studies that have included use of the FGD on sensitive topics or with vulnerable groups i.e sex workers. We believe that parallels can be drawn with our study and should reassure Reviewer 2 that sensitive information can be obtained through the focus group discussion method, assuming that all safeguards and ethical procedures are followed correctly. 

• McGowan, M., Roche, S.D., Nakitende, A. et al. Understanding how social support influences peer-delivered HIV prevention interventions among Ugandan female sex workers: a case study from HIV self-testing. BMC Public Health 22, 427 (2022). https://doi.org/10.1186/s12889-022-12836-3

• Sikhosana N, Mokgatle MM. A qualitative exploration on accounts of condom-use negotiation with clients: challenges and predicaments related to sex work among street-based female sex workers in Ekurhuleni District, South Africa. Pan Afr Med J. 2021 Sep 22;40:54. doi: 10.11604/pamj.2021.40.54.29918. PMID: 35059100; PMCID: PMC8724014.

• Crankshaw TL, Chareka S, Zambezi P, Poku NK. Age Matters: Determinants of s

---

## [Decision Letter · Decision Letter 1]

13 Nov 2024

PONE-D-24-10017R1'Once they see blood then the mood for sex is spoiled’ A Qualitative Exploration of Female Sex Worker’s Male Client Views of Menstruation, Sex during Menses and The Menstrual DiscPLOS ONE

Dear Dr. Mason,

Thank you for submitting your manuscript to PLOS ONE. After careful consideration, we feel that it has merit but does not fully meet PLOS ONE’s publication criteria as it currently stands. Therefore, we invite you to submit a revised version of the manuscript that addresses the points raised during the review process.

Please address the minor comments from Reviewer 3 in your revised manuscript. Once these have been addressed your manuscript will be suitable for publication subject to final journal requirements. 

 Please submit your revised manuscript by Dec 21 2024 11:59PM. If you will need more time than this to complete your revisions, please reply to this message or contact the journal office at plosone@plos.org. Please include the following items when submitting your revised manuscript:A rebuttal letter that responds to each point raised by the academic editor and reviewer(s). You should upload this letter as a separate file labeled 'Response to Reviewers'.A marked-up copy of your manuscript that highlights changes made to the original version. You should upload this as a separate file labeled 'Revised Manuscript with Track Changes'.An unmarked version of your revised paper without tracked changes. You should upload this as a separate file labeled 'Manuscript'.If applicable, we recommend that you deposit your laboratory protocols in protocols.io to enhance the reproducibility of your results. Protocols.io assigns your protocol its own identifier (DOI) so that it can be cited independently in the future. For instructions see: https://journals.plos.org/plosone/s/submission-guidelines#loc-laboratory-protocols. Additionally, PLOS ONE offers an option for publishing peer-reviewed Lab Protocol articles, which describe protocols hosted on protocols.io. Read more information on sharing protocols at https://plos.org/protocols?utm_medium=editorial-email&utm_source=authorletters&utm_campaign=protocols.

We look forward to receiving your revised manuscript.

Kind regards,

Emma Campbell, Ph.D

Staff Editor

PLOS ONE

on behalf of 

Martin Mbonye

Academic Editor

PLOS ONE

Journal Requirements:

Reviewers' comments:

Reviewer's Responses to Questions

**Comments to the Author**

1. If the authors have adequately addressed your comments raised in a previous round of review and you feel that this manuscript is now acceptable for publication, you may indicate that here to bypass the “Comments to the Author” section, enter your conflict of interest statement in the “Confidential to Editor” section, and submit your "Accept" recommendation.

Reviewer #3: All comments have been addressed

2. Is the manuscript technically sound, and do the data support the conclusions?

Reviewer #3: Yes

3. Has the statistical analysis been performed appropriately and rigorously? 

Reviewer #3: N/A

4. Have the authors made all data underlying the findings in their manuscript fully available?

Reviewer #3: Yes

5. Is the manuscript presented in an intelligible fashion and written in standard English?

Reviewer #3: Yes

6. Review Comments to the Author

Reviewer #3: I have reviewed the responses to comments and the manuscript. The authors have responded carefully to all the comments from both reviewers. In particular, for reviewer #1, the authors have added additional detail which clearly articulates the gap and links this through the results and discussion. For reviewer #2, the authors have added additional detail and supporting references, to explain their use of FGDs for this topic and population. From my reading, the use of FGDs was strongly informed by ethical concerns and practices, and the authors (and facilitator) approach was grounded in contextual understanding of the issue and challenges. I do not share reviewer #2 concerns.

I only have two minor comments:

Line : 122 - page 6.

“this method allows group dynamics…” - Maybe just a rephrase is needed here to emphasise that the group dynamics enabled (the bonding experience)

page 26 - conclusion:

Would it be possible to add an extra sentence or even just a few words linking back to the focus on FSW’s male client views of menstruation? The introduction, results, and discussion clearly articulate the intersecting emphasis/gap/significance. It would be great to see it mirrored more here.

7. PLOS authors have the option to publish the peer review history of their article (what does this mean?). If published, this will include your full peer review and any attached files.

Reviewer #3: No

---

## [Author Response · Author response to Decision Letter 1]

18 Nov 2024

‘once they see blood then the mood for sex is spoiled’ A Qualitative Exploration of Female Sex Worker’s Male Client Views of Menstruation, Sex during Menses and The Menstrual Disc

Dear Editor and Reviewer 3,

Thank you very much for taking the time and effort to review our paper along with the previous reviewer feedback. We have outlined below the changes we have made in accordance with both of your suggestions. We hope that our amendments meet with your approval. 

Line : 122 - page 6.

“this method allows group dynamics…” - Maybe just a rephrase is needed here to emphasise that the group dynamics enabled (the bonding experience)

Thank you for your suggestion. We have amended as per below:

‘This method can facilitate a productive discussion through group dynamics, and lead to a bonding experience for participants who we hypothesised, may have felt embarrassed or reticent in talking about menstruation in a one-to-one situation with a researcher.’

page 26 - conclusion:

Would it be possible to add an extra sentence or even just a few words linking back to the focus on FSW’s male client views of menstruation? The introduction, results, and discussion clearly articulate the intersecting emphasis/gap/significance. It would be great to see it mirrored more here.

We hope the following amendments have been able to link our conclusions back to the focus as suggested. The conclusion now reads:

‘Our study has added to the limited research on men’s knowledge and attitudes toward menstruation focusing on their views of sex during menses with FSW to help understand how it may contribute to unsafe or risky menstrual hygiene practices that these women undertake. This research highlights men’s lack of knowledge and largely negative perceptions about the topic, including towards sex during menses. Researchers, policy makers, politicians alike have been calling for increased awareness of boys and men around menstruation. We echo this need, calling for accurate information including on school curricula, and through faith teaching where appropriate, to address the current lack of awareness that contributes to stigma and shame, and perpetuation of menstrual taboos. This research illuminates men’s need for education around reproductive health in general, noting that participants were receptive to, and curious about a reusable menstrual disc that can be worn during sex, which suggests that this has potential to be adopted by FSW with minimal adverse client reaction. Further it highlights possible partner support for women considering adopting a menstrual disc in the future. These study findings can be used to inform successful menstrual health programme implementation for FSW in the future’.

---

## [Decision Letter · Decision Letter 2]

26 Nov 2024

'Once they see blood then the mood for sex is spoiled’ A Qualitative Exploration of Female Sex Worker’s Male Client Views of Menstruation, Sex during Menses and The Menstrual Disc

PONE-D-24-10017R2

Dear Dr. Mason,

We’re pleased to inform you that your manuscript has been judged scientifically suitable for publication and will be formally accepted for publication once it meets all outstanding technical requirements.

Kind regards,

Alison Parker

Academic Editor

PLOS ONE

Additional Editor Comments (optional):

Reviewers' comments:

Reviewer's Responses to Questions

**Comments to the Author**

1. If the authors have adequately addressed your comments raised in a previous round of review and you feel that this manuscript is now acceptable for publication, you may indicate that here to bypass the “Comments to the Author” section, enter your conflict of interest statement in the “Confidential to Editor” section, and submit your "Accept" recommendation.

Reviewer #3: All comments have been addressed

2. Is the manuscript technically sound, and do the data support the conclusions?

Reviewer #3: Yes

3. Has the statistical analysis been performed appropriately and rigorously? 

Reviewer #3: N/A

4. Have the authors made all data underlying the findings in their manuscript fully available?

Reviewer #3: Yes

5. Is the manuscript presented in an intelligible fashion and written in standard English?

Reviewer #3: Yes

6. Review Comments to the Author

Reviewer #3: The authors have addressed the comments and made thoughtful adjustments to the methods and conclusion.

The conclusion now clearly reflects the focus on FSW.

7. PLOS authors have the option to publish the peer review history of their article (what does this mean?). If published, this will include your full peer review and any attached files.

Reviewer #3: No

---

## [Editor Report · Acceptance letter]

13 Dec 2024

PONE-D-24-10017R2 

PLOS ONE

Dear Dr. Mason, 

I'm pleased to inform you that your manuscript has been deemed suitable for publication in PLOS ONE. Congratulations! Your manuscript is now being handed over to our production team.

Kind regards, 

on behalf of

Dr. Alison Parker 

Academic Editor

PLOS ONE